# Enhanced Face Recognition using Intra-class Incoherence

**Yuanqing Huang**[1][*]  **Yinggui Wang**[1][*][†]  **Le Yang**[2]  **Lei Wang**[1]
Ant Group[1]    University of Canterbury[2]
{huangyuanqing.hyq, yinggui.wyg, shensi.wl}@antgroup.com    le.yang@canterbury.ac.nz

## Abstract

The current face recognition (FR) algorithms has achieved a high level of accuracy, making further improvements increasingly challenging. While existing FR algorithms primarily focus on optimizing margins and loss functions, limited attention has been given to exploring the feature representation space. Therefore, this paper endeavors to improve FR performance in the view of feature representation space. Firstly, we consider two FR models that exhibit distinct performance discrepancies, where one model exhibits superior recognition accuracy compared to the other. We implement orthogonal decomposition on the features from the superior model along those from the inferior model and obtain two sub-features. Surprisingly, we find the sub-feature orthogonal to the inferior still possesses a certain level of face distinguishability. We adjust the modulus of the sub-features and recombine them through vector addition. Experiments demonstrate this recombination is likely to contribute to an improved facial feature representation, even better than features from the original superior model. Motivated by this discovery, we further consider how to improve FR accuracy when there is only one FR model available. Inspired by knowledge distillation, we incorporate the intra-class incoherence constraint (IIC) to solve the problem. Experiments on various FR benchmarks show the existing state-of-the-art method with IIC can be further improved, highlighting its potential to further enhance FR performance.

## 1 Introduction

Face recognition (FR) has been receiving increasing attention in the field of machine learning. With the development of deep learning and the availability of large-scale face datasets, the accuracy of face recognition tasks is steadily improving. Many FR algorithms are devoted to optimizing margins and loss functions. Margin-based methods such as SphereFace (Liu et al., 2017), CosFace (Wang et al., 2018b) and ArcFace (Deng et al., 2019) introduce different forms of margin functions. Adaptive loss functions-based methods such as Curricularface (Huang et al., 2020b), MagFace (Meng et al., 2021) and AdaFace (Kim et al., 2022) include adaptiveness in the training objective for mining hard samples (Lin et al., 2017), scheduling difficulty during training (Huang et al., 2020b), and finding optimal hyperparameters (Zhang et al., 2019). Nevertheless, the exploration of the feature representation space in face recognition has been somehow overlooked. From the perspective of feature representation, an advanced FR model makes the features of face images increasingly distinguishable and attempts to search the optimal feature space with the most representational capability for recognition.

Therefore, we are absorbed in interpreting the performace of FR from the view of the feature space and further considering how to enhance the performance of an existing face recognition model. Firstly, we compare two FR models that exhibit distinct performance discrepancies. As shown in Fig. 1(a), for the clarity of presentation, we use two-dimensional vectors to illustrate the feature representations space of different FR models at the feature level, and choose ArcFace (Deng et al., 2019) and CosFace (Wang et al., 2018b) as examples. Here, $d$ represents an optimal but unknown feature representation space for FR. $a$ and $b$ represent the feature space of ArcFace and CosFace.

---

[*]These authors contributed equally to this work.
[†]Corresponding author.

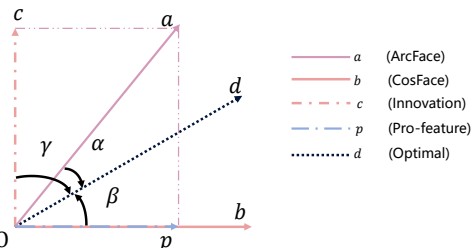
(a) $a$ and $b$ represent the features from two face recognition models (such as ArcFace and CosFace). $d$ is an optimal but *unknown* facial feature. We obtain two sub-features $c$ and $p$ (innovation and pro-feature) from $a$ by orthogonal decomposition. $p$ is parallel to $b$, and $c$ is orthogonal to $b$.

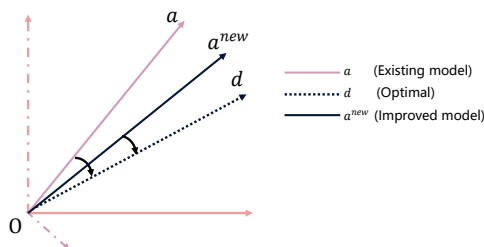
(b) For an existing model feature $a$, if innovation (i.e. $a^\perp$) is able to be learned, the improved model feature $a^{new}$ can be obtained by combining innovation and $a$.

Figure 1: Orthogonal decomposition in the feature space of face recognition models.

Table 1: The recognition accuracy of features from ArcFace, CosFace, innovation and pro-feature decomposed by ArcFace, respectively.

| Dateset | CosFace | ArcFace | Innovation | Pro-feature |
|---|---|---|---|---|
| LFW | **99.22** | 99.15 | 99.17 | 86.32 |
| CFP | 93.87 | 95.69 | **95.86** | 79.66 |
| AgeDB | 92.55 | **94.35** | 92.97 | 79.73 |
| CALFW | 92.77 | **93.05** | 92.58 | 80.98 |
| CPLFW | 87.57 | **88.20** | 88.00 | 76.17 |
| VGG2 | 91.94 | **92.66** | 91.98 | 77.66 |
| AVG (Average accuracy) | 92.99 | **93.85** | 93.19 | 80.09 |

Since it is generally believed that ArcFace perform better than CosFace, $a$ should be closer to $d$, compared with $b$ (i.e. $\angle\alpha < \angle\beta$). We perform orthogonal decomposition on the feature obtained from ArcFace and get two sub-features, where one of the sub-features $p$ is parallel to $b$ (we call the sub-feature **pro-feature**). The other sub-feature $c$ is orthogonal to $b$ (we call it **innovation**).

In Table 1, we evaluate FR performance over features from ArcFace, CosFace, innovation and pro-feature decomposed by ArcFace, respectively (depicted as in Fig. 1(a)). The results show that ArcFace has the best performance in most cases, while innovation exhibits certain capabilities in distinguishing faces since its average accuracy outperforms CosFace. Unexpectedly, the accuracy of pro-feature is far lower than that of CosFace. The reasons behind this may be explained as follows. The angle between the feature from ArcFace and the feature from CosFace may be less than 90 degrees in some cases and more than 90 degrees in other cases. When it is less than 90 degrees, the direction of the pro-feature is exactly aligned with that from CosFace, and the recognition accuracy is thus similar. While it is greater than 90 degrees, the direction of the pro-feature would become opposite to the direction of features from CosFace, rendering poor recognition accuracy and degrading the average FR performance. However, it does not influence the accuracy of innovation since innovation is always orthogonal to features from CosFace. Such discovery can lead to a conclusion that the high-precision model (ArcFace) performs better than the low-precision model (CosFace) because the former contains innovations that are irrelevant to the feature from the low-precision model, while these innovations do have a promoting effect on face recognition. In other words, we can optimize the features extracted from an available FR model by incorporating appropriate innovations to enhance its performance.

With the above analysis in mind, innovation motivates us to improve FR accuracy in the scenario where only a single FR model is available, as shown in Fig. 1(b). We attempt to utilize intra-class irrelevance on the feature from an existing state-of-the-art model such that its innovation can be learned (denoted by $a^\perp$ in Fig. 1(b)), properly scaled, and added to its original feature to get an improved feature representation space(denoted by $a^{new}$ in Fig. 1(b)). Consequently, the improved feature representation would be further closer to the optimal feature $d$.

In general, the contributions of this work are as follows:

- The concept of innovation in FR is put forward for the first time. Considering two FR models, we perform orthogonal decomposition on the features from the superior model with higher recognition accuracy. The sub-feature parallel to the features from the inferior model is **pro-feature**, and the other sub-feature orthogonal to features from the inferior model is **innovation**. Experiments show that innovation still has a high level of face distinguishability.

- Moreover, innovation inherently has a positive impact on face recognition. We adjust the modulus of innovation, and then synthesize new features by recombining the modified innovation and pro-feature. Experiments indicate this synthesized features are likely to have better representational capability for recognition than the original feature.

- Furthermore, we consider how to enhance FR performance when there is only one FR model available. Inspired by knowledge distillation, we adopt a framework incorporating the intra-class irrelevance constraint. In our framework, an existing state-of-the-art FR model serves as the teacher network and the student has the same architecture as the teacher. However, the student network not only possesses the capability for recognition but also learns a **dissimilar** feature representation space. Experiments show that the performance of the student network outperforms the teacher network by incorporating intra-class irrelevance.

## 2 RELATED WORK

### 2.1 FACE RECOGNITION

Face recognition algorithms are generally divided into methods with metric-learning-based loss functions, methods with adaptive loss functions and those with uncertainty modeling. Early methods with metric learning include contrastive loss (Liu et al., 2021), triplet loss (Schroff et al., 2015), pair loss (Sohn, 2016) and angular loss (Wang et al., 2017). Then, in this category, some algorithms with higher accuracy have been proposed, which include SphereFace (Liu et al., 2017), AM-softmax (Wang et al., 2018a), SV-AM-Softmax (Wang et al., 2018c), CosFace (Wang et al., 2018b), ArcFace (Deng et al., 2019).

Methods with adaptive loss functions introduce an element of adaptiveness in the training objective for either hard sample mining (Lin et al., 2017), finding optimal hyperparameters (Zhang et al., 2019), or brings the idea of curriculum learning into the loss function (Huang et al., 2020b). In this category, MagFace (Meng et al., 2021) and AdaFace (Kim et al., 2022) are algorithms with higher accuracy. MagFace explores the idea of applying different margins based on recognizability. It applies large angular margins to high-norm features on the premise that high-norm features are easily recognizable. AdaFace develops a new loss function that emphasizes samples of different difficulties based on their image quality, and it achieves this in the form of an adaptive margin function by approximating the image quality with feature norms. Algorithms with uncertainty modeling include probabilistic face embeddings (PFE (Shi & Jain, 2019)), sphere confidence face (SCF (Li et al., 2021)), and data uncertainty learning in FR (DUL (Chang et al., 2020)). PFE is the first work to consider data uncertainty in FR, and it estimates a Gaussian distribution, instead of a fixed point, in the latent space. While being effective, PFE is limited in that it does not learn the embedded feature (mean) but only the uncertainty. DUL applies data uncertainty learning (DUL) to face recognition such that feature (mean) and uncertainty (variance) are learned simultaneously. Besides, SCF theoretically and empirically identifies two main failures of PFE when it is applied to spherical deterministic embeddings aforementioned. To address these issues, SCF develop a novel framework for face confidence learning in spherical space, and extend the von Mises Fisher density to its r-radius counterpart and derive a new optimization objective in closed form.

Terhörst et al. (2020) introduced a seemingly similar concept, called "negative face representation". However, this method was primarily focused on privacy protection, as its "negative face representation" solely influenced the face verification stage and did not impact the embedding mapping process where facial features are extracted. Furthermore, it did not contribute to improving the recognition accuracy of the FR model. In contrast, our method directly influences the face-to-embedding mapping process, leading to an improvement in the recognition accuracy of the final model.

## 2.2 KNOWLEDGE DISTILLATION

The method in this paper uses the framework of knowledge distillation. The general methods of knowledge distillation and the methods used in face recognition knowledge distillation are briefly introduced below.

Knowledge distillation has been actively investigated and widely used in many computer vision tasks. The basic idea proposed by Hinton et al. (2015) minimizes the KL divergence of softened class probabilities between the teacher and the student. Later variants of distillation strategies are proposed to utilize diverse information from the teacher model, which can be divided into two types, i.e., logit distillation and feature distillation. As for the feature distillation, typical algorithms include FitNet (Romero et al., 2014), ShrinkTeaNet (Duong et al., 2019), TripletDistillation (Feng et al., 2020), and MarginDistillation (Svitov & Alyamkin, 2020). Different from feature-based distillation methods, logit distillation does not require the student to mimic the teacher's representation space, but rather to preserve the high semantic consistency with the teacher. The classical methods include the Kullback-Leibler Divergence-based algorithm (Hinton et al., 2015), the mutual learning based algorithm (Zhang et al., 2018), and DKD (Zhao et al., 2022), which firstly decouple the classical KD Loss into target class knowledge distillation and non-target class knowledge distillation.

The typical methods used in face recognition knowledge distillation include evaluation-oriented knowledge distillation for FR (EKD) (Huang et al., 2022), and improved FR via distribution distillation loss (DDL) (Huang et al., 2020a). EKD introduces the evaluation-oriented method that optimizes the student model's critical relations that cause the TPR and FPR difference between the teacher and student models. DDL uses the distillation loss to minimize the gap between easy and hard sample features. A recent research Shin et al. (2022) introduced attention similarity by a knowledge distillation framework where both the student and teacher networks are of the same size. However, it still follows a traditional paradigm of knowledge distillation, attempting the student network to learn knowledge that is similar to the teacher network. In contrast, our proposed method aims for the student network to learn different knowledge from the teacher network.

## 3 PROPOSED APPROACH

The fundamental goal of FR algorithms is to learn the optimal feature space with the most representational capability for recognition. In this paper, we suppose the feature representation space change reflects in the accuracy improvement. Firstly, we provide a detailed analysis of the feature space by feature decomposition and recombination in Section 3.1. Experimental results show that such recombination is likely to contribute to an improved facial feature space. However, it is a challenge to find innovation in a scenario where there is only one single model available. To solve the problem, we adopt a framework inspired by knowledge distillation in Section 3.1. We utilize intra-class incoherence constraint (IIC) to learn innovation and finally obtain a better facial feature representation from an existing model.

### 3.1 FEATURE DECOMPOSITION AND RECOMBINATION

Firstly, we review the feature decomposition in Introduction, and compare two FR models that exhibit distinct performance discrepancies. We perform orthogonal decomposition on the feature from the superior model and obtain two sub-features, pro-feature and innovation. The pro-feature is parallel to the feature from the inferior model, and innovation is orthogonal to the feature from the inferior model. Let $a$ be the feature extracted from the superior model, $b$ be the feature obtained by the inferior model, and $c$ be innovation. According to the Gram-Schmidt Orthogonalization, we have

$$c = a - \frac{\langle a, b \rangle}{\|b\| \|b\|} b, \tag{1}$$

where $\langle a, b \rangle$ denotes the inner product operation of $a$ and $b$, $\| \cdot \|$ represents the $L2$-norm.

Table 1 indicates that innovation also has a certain level of face distinguishability. This indicates that the feature $a$ indeed contains useful information irrelevant to the feature $b$. In other words, with the help of innovation, the feature $a$ gets closer to the optimal direction $d$. Since the optimal feature direction is not known, it is natural to investigate whether we can change the modulus of innovation,

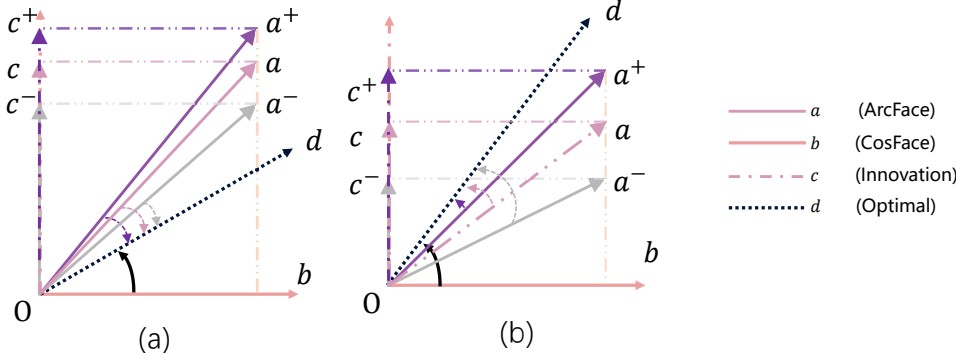

Figure 2: The schematic diagrams of synthesizing new features under different innovation modulus settings. (a) $a$ and $b$ are on different sides of $d$, and (b) $a$ and $b$ are on the same side of $d$. $c^+$ and $c^-$ represent the increase and decrease of $c$'s modulus. $a^+$ and $a^-$ denote the feature representation newly synthesized based on $c^+$ and $c^-$.

and synthesize a new feature $a^{new}$ using modified innovation and pro-feature, with the hope that $a^{new}$ (depicted in Fig. 1(b)) would be closer to the optimal direction than $a$.

According to Fig. 2, there are two cases for synthesizing features. In the first case, as shown in Fig. 2(a), $a$ and $b$ are on different sides of $d$. In this case, it is necessary to reduce the modulus of innovation (or increase the modulus of pro-feature), so that $a^{new}$ can be closer to the direction $d$. However, excessive reduction of the modulus of innovation (or increase the modulus of pro-feature too much) will lead to the angle between $a^{new}$ ($a^-$ or $a^+$) and $d$ being greater than that between $a$ and $d$, which will eventually impact the recognition accuracy. In the second case, as shown in Fig. 2(b), $a$ and $b$ are on the same sides of $d$. In this case, it is necessary to increase the modulus of innovation (or reduce the modulus of pro-feature), so that $a^{new}$ may be closer to the direction $d$. Similarly, extra care is required when increasing the modulus. We take an example where ArcFace servers as $a$ and CosFace servers as $b$, and show the performance of the above recombinations in Table 2. We can observe that this recombination is likely to contribute to an improved facial feature representation, even better than features from ArcFace.

Motivated by this discovery, we further investigate a more practical scenario where only a single model is available (like $a$ in Fig. 1(b)), but its improved model ($a^{new}$ in Fig. 1(b)) and innovation ($a^\perp$ Fig. 1(b)) are unknown. We need to search for innovation from the existing feature space and properly scale it to synthesize a better feature. Through the aforementioned analysis, we identify two key characteristics of innovation. Firstly, the innovation $a^\perp$ is orthogonal to $a$. Secondly, it facilitates an improvement in recognition accuracy.

Table 2: The recognition accuracy of recombined features under different innovation modulus settings. Innovation and pro-feature are sub-features of ArcFace. A novel feature can be obtained by combing them with '$c*$Innovation + Pro-feature', where $c$ is the coefficient. This table shows the performance of various features under different coefficients (the feature exactly represents ArcFace when $c = 1$).

| Dateset | ArcFace | 0.1 | 0.5 | 0.8 | 0.9 | 1.1 | 1.2 | 2.0 | 5.0 |
|---|---|---|---|---|---|---|---|---|---|
| LFW | 99.15 | 98.82 | 99.20 | 99.12 | 99.13 | 99.15 | 99.15 | 99.17 | 99.17 |
| CFP | 95.69 | 94.77 | 95.61 | 95.67 | 95.61 | 95.69 | 95.73 | 95.73 | 95.70 |
| AgeDB | 94.35 | 92.97 | 94.05 | 94.37 | 94.35 | 94.37 | 94.37 | 94.13 | 94.13 |
| CALFW | 93.05 | 92.58 | 93.03 | 93.00 | 94.03 | 93.05 | 93.03 | 92.88 | 92.88 |
| CPLFW | 88.20 | 88.00 | 88.45 | 88.27 | 88.22 | 88.47 | 88.43 | 88.22 | 88.23 |
| VGG2 | 92.66 | 91.98 | 92.74 | 92.66 | 92.68 | 92.66 | 92.66 | 92.66 | 92.62 |
| AVG | 93.85 | 93.19 | 93.85 | 93.85 | **94.00** | 93.90 | 93.90 | 93.80 | 93.79 |

## 3.2 FR WITH INTRA-CLASS INCOHERENCE

Our purpose is to learn the superior feature containing innovation from an existing model feature. Based on the aforementioned observations and analysises, we propose a novel FR training paradigm incorporating intra-class incoherence.

Specifically, we first consider utilizing the knowledge distillation(KD) framework to implement intra-class incoherence. For traditional KD, the teacher network is a model trained with the face recognition algorithm with better performance. The student network is a network that needs to be trained. The teacher network supervises the student network at the feature and logit levels. However, due to the different network structures of the teacher and student networks, the feature dimensions are different, and the supervision on the feature level sometimes is optional.

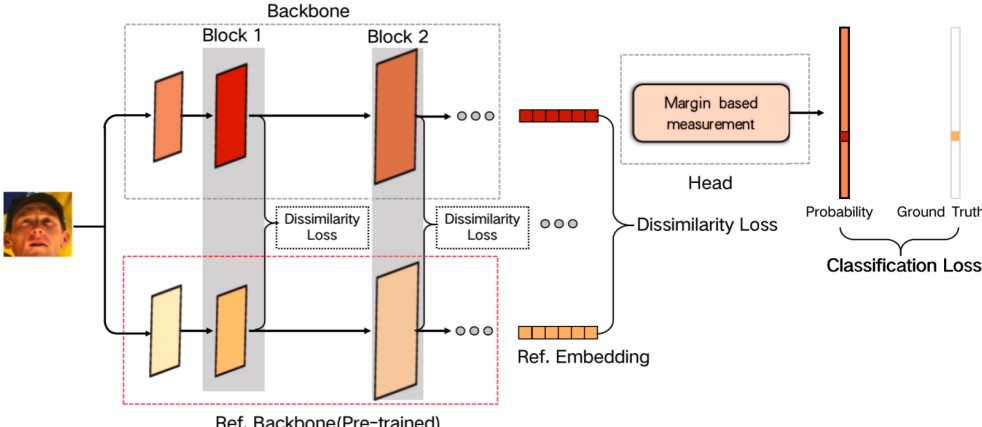

Figure 3: Schematic diagram of imposing intra-class irrelevant constraints (IIC) on output features of different layers of the feature extraction network.

Nevertheless, different from the traditional knowledge distillation methods, the student network in our framework keeps the same size as the teacher to avoid impairing the ability of recognition. As shown in Fig 3, the two models are the same in architecture, and we expect the to-be-trained model (the student network) to perform better than the pre-trained model (the teacher network). Another noteworthy difference is that the general KD is to learn the relevant knowledge of a teacher network, while our method obtains the irrelevant information of a teacher network with intra-class irrelevant constraints (i.e. feature dissimilarity).

Therefore, the loss function $L_S$ for student network training is composed of the face recognition loss function $L_{FR}$ and the intra-class incoherence cost $L_{dissim}$.

$$L_S = L_{FR} + \gamma L_{dissim}, \tag{2}$$

where $\gamma$ is the weight, $L_{dissim}$ is the cosine similarity of $f_T$ and $f_S$, and $f_T$ and $f_S$ are the feature embeddings of the same face image obtained through the teacher and student networks. The term $L_{FR}$ is to preserve the capability for face recognition and $L_{dissim}$ is to impose IIC.

To introduce innovation of different levels when training, we take such a strategy: in addition to the final feature embeddings of the feature extraction network, the proposed intra-class irrelevance is also applied to different intermediate feature levels. Specifically, we take IR-SE-50 as the feature extraction network for experiments. As shown in Fig. 3, IR-SE-50 also has four blocks. We investigate updating the four blocks' outputs using intra-class incoherence via

$$L_S = L_{FR} + \gamma_i . L_{dissim_i}, (i = 1, \ldots, N), \tag{3}$$

where $\gamma_i$ is the weight, $L_{dissim_i}$ is the cosine similarity between $f_{T_i}$ and $f_{S_i}$, $f_{T_i}$ and $f_{S_i}$ are the features of the same face image obtained through the $i$th block of the teacher network and student network, $N$ denotes the number of blocks in the feature extraction network.

We also consider the case where the intra-class incoherence is applied to the outputs of the four modules and the final output embedding of the feature extraction network in the following hybrid

way:

$$L_S = L_{FR} + \sum_{i=1}^{N+1} \gamma_i . L_{dissim_i}, \tag{4}$$

where $N+1$ denotes the number of blocks and the output layer in the feature extraction network. By controlling whether $\gamma_i$ is zero, we can examine the effect of intra-class incoherence constraints over an arbitrary combination of different output layers on FR accuracy. See Section 4.3 for detailed ablation studies and hyper-parameter preference.

# 4 EXPERIMENTS

## 4.1 DATASETS AND IMPLEMENTATION DETAILS

We use CASIA (Yi et al., 2014) and MS1M-ArcFace (also known as MS1MV2) (Guo et al., 2016) as the training datasets. 7 common benchmarks including LFW (Zhang & Deng, 2016), CFP-FP (Sengupta et al., 2016), CPLFW(Zhang & Deng, 2016), AgeDB (Moschoglou et al., 2017), CALFW (Zhang & Deng, 2016), Vggface2 (Cao et al., 2018), and IJB-C (Maze et al., 2018) are used to evaluate the performance of different algorithms following the standard evaluation protocols. For these test sets, the images show variations in lighting, pose, or age.

We train the model with ResNet50 and ResNet100 (He et al., 2016) as the backbone and batch size of 512 using the metric and loss functions similar to the specified definition in their original text. The head of the baseline model is: BackBone-Flatten-FC-BN with embedding dimensions of 512 and the dropout probability of 0.4 to output the embedding feature. Unless specified otherwise, models are trained for 50 epochs using the SGD optimizer with a momentum of 0.9, and a weight decay of 0.0001. The model is trained with SGD with an initial learning rate of 0.1 and step scheduling at 10, 20, 30 and 40 epochs. For the scale parameters, we set it to 64, following the suggestion of Wang et al. (2018b).

## 4.2 COMPARISON WITH SoTA METHODS AND ANALYSIS

In this experiment, we adopt some common baseline models such as ArcFace (Deng et al., 2019) and CosFace (Wang et al., 2018b). Besides, some recent state-of-the-art methods such as MagFace (Meng et al., 2021) and AdaFace (Kim et al., 2022) also serve as the baseline models for comparison, and this paper adopts the training paradigm with IIC in a manner consistent with the original literature. Since the proposed method is based on a pre-trained model, the parameters of the teacher network are downloaded from their official offered checkpoints if available.

Firstly, we train a relatively small network with ResNet50 (He et al., 2016) as the backbone on CASIA (Yi et al., 2014). The experimental results are shown in Table 3.

Table 3: Comparison of face recognition accuracy among some common and recent FR algorithms. All the models are trained with ResNet50 (He et al., 2016) as the backbone on CASIA.

| Method | Benchmarks(%) | | | | | | Average |
|---|---|---|---|---|---|---|---|
| | LFW | CFP-FP | AgeDB-30 | CALFW | CPLFW | VGGFace2 | |
| CosFace | 99.22 | 93.87 | 92.55 | 92.77 | 87.57 | 91.94 | 92.99 |
| CosFace+IIC | 99.17 | 94.40 | 92.48 | 92.53 | 88.22 | 91.94 | **93.12** |
| ArcFace | 99.15 | 95.69 | 94.35 | 93.05 | 88.20 | 92.66 | 93.85 |
| ArcFace+IIC | 99.43 | 96.03 | 94.12 | 93.05 | 89.45 | 93.36 | **94.24** |
| MagFace | 99.27 | 95.27 | 93.77 | 93.33 | 87.90 | 92.10 | 93.61 |
| MagFace+IIC | 99.25 | 95.70 | 93.58 | 92.95 | 88.08 | 92.58 | **93.69** |
| AdaFace | 99.42 | 96.41 | 94.38 | 93.23 | 89.97 | 93.08 | 94.42 |
| AdaFace+IIC | 99.18 | 96.94 | 94.10 | 93.37 | 90.17 | 93.74 | **94.58** |

We observe that IIC proposed in this paper has significant improvements on all face recognition algorithms. Even the recognition accuracy of the SoTA method AdaFace is also improved. By comparing the results of Table 2 and Table 3 (results of ArcFace), it is evident that the improvement in recognition accuracy achieved by IIC surpasses that listed in Table 2. Then we are curious about

the performance of IIC on some large-scale datasets and more complex networks. Therefore, we train a ResNet100 network as the backbone using MS1MV2 as the training dataset and show the results in Table 4.

Table 4: Comparison of face recognition accuracy among state-of-the-art FR algorithms. All the models are trained with ResNet100 as the backbone on MS1MV2.

| Method | Benchmarks(%) | | | | | | Average |
|---|---|---|---|---|---|---|---|
| | LFW | CFP-FP | AgeDB-30 | CALFW | CPLFW | VGGFace2 | |
| CosFace | 99.78 | 98.12 | 98.11 | 95.76 | 92.18 | 95.24 | 96.53 |
| CosFace+IIC | 99.80 | 98.24 | 98.16 | 95.68 | 92.26 | 95.34 | **96.58** |
| ArcFace | 99.78 | 98.10 | 98.20 | 95.87 | 92.62 | 95.74 | 96.72 |
| ArcFace+IIC | 99.78 | 98.19 | 98.25 | 96.05 | 93.02 | 95.30 | **96.76** |
| MagFace | 99.83 | 98.46 | 98.17 | 96.15 | 92.87 | 95.40 | **96.81** |
| MagFace+IIC | 99.82 | 98.14 | 98.12 | 96.17 | 92.88 | 95.40 | 96.76 |
| AdaFace | 99.82 | 98.49 | 98.05 | 96.08 | 93.53 | 95.70 | 96.94 |
| AdaFace+IIC | 99.78 | 98.41 | 98.12 | 96.18 | 93.48 | 95.80 | **96.96** |

Table 5: Comparison of the face recognition accuracy for the ArcFace and AdaFace algorithms before and after using IIC. The training dataset is MS1M, and the test dataset is IJB-C.

| Method (Backbone) | IJB-C (TPR@FPR) | | | | | |
|---|---|---|---|---|---|---|
| | 1e-06 | 1e-05 | 1e-04 | 1e-03 | 1e-02 | 1e-01 |
| ArcFace(R50) | 85.64 | 92.88 | 95.92 | 97.45 | 98.44 | 99.21 |
| ArcFace+IIC(R50) | **89.50** | **94.53** | **96.28** | **97.52** | **98.48** | **99.23** |
| MagFace(R50) | 81.70 | 88.95 | 93.34 | 95.91 | 97.70 | 99.01 |
| MagFace+IIC(R50) | **85.88** | **92.37** | **95.16** | **96.94** | **98.15** | **99.06** |
| AdaFace(R50) | 87.70 | 93.61 | 96.27 | 97.62 | 98.29 | 99.10 |
| AdaFace+IIC(R50) | **87.87** | **94.06** | **96.33** | **97.64** | **98.38** | **99.12** |
| MagFace(R100) | 89.26 | 93.67 | 95.81 | 97.26 | **98.27** | 99.10 |
| MagFace+IIC(R100) | **89.38** | **93.95** | **95.89** | **97.28** | 98.27 | **99.12** |
| AdaFace (R100) | 89.74 | 94.87 | **96.89** | **97.89** | 98.47 | **99.18** |
| AdaFace+IIC (R100) | **89.99** | **94.88** | **96.89** | **97.89** | **98.51** | 99.17 |

To provide a comprehensive evaluation, we present the performance of state-of-the-art (SoTA) methods on the IJB-C benchmark in Table 5. In this setting, we utilize the MS1MV2 dataset for training. From the results in Table 5, it is evident that our method outperforms the baseline models in most cases, particularly when a low false positive rate (FPR) is required. Notably, when the FPR is set 1e-6, the models trained with the intra-class incoherence constraint (IIC) significantly outperform all baseline methods.

However, comparing the results in Tab 4 and Tab 3, we find that IIC plays a more effective role in a relatively smaller dataset. This observation motivates us to investigate the underlying reasons. Through careful analysis of the experimental results, we associate it with the technique of data augmentation. Data augmentation is a technique of artificially increasing the training set by creating modified copies of a dataset using existing data. It introduces perturbations to the data, making it challenging for the model to simply recognize them. Although data augmentation may seemingly increase the training difficulty, it actually enables the model to better learn the underlying feature representations. Furthermore, data augmentation has been shown to effectively enhance the performance of models when dealing with limited data. These explanations can also be applied to IIC. Our proposed method encourages the FR model to explore a different feature space and learn a better feature representation. Therefore, we hypothesize that IIC serves as a form of "feature augmentation". To validate this hypothesis, we conduct an ablation study in Sec 4.3.

### 4.3 ABLATION STUDY AND HYPER-PARAMETER PREFERENCE

In this section, if not specified, we use the CASIA dataset to train the model in which ArcFace serves as the basic face recognition algorithm.

Firstly, we investigate the impact of different weights on the performance of Eq. 2. The detailed results are shown in Appendix Table 6. We find that the average precision on all test sets is improved. For different weights in Table 6, the performance gap is marginal. For convenience, the previous experiments also have a similar treatment. In the absence of special instructions, the default weight is 1.0.

Besides, we examine the impact of the different initialization methods for student networks on recognition accuracy, namely, random initialization and initialization with the teacher network's weight. We provide the results in Appendix Table 7. It is obvious that the recognition accuracy of initialization with the teacher network's weight is the best. Unless otherwise specified, the weight of the teacher network is used for initialization by default. We also investigate the case where the student network has no intra-class irrelevant constraints, that is, only the weight of the teacher network is loaded for initialization and the model is trained solely by FR loss. It can be seen from Table 7 that the absence of intra-class irrelevant constraints does not improve the performance, but the performance will decline slightly. We suppose it is because that the data are reused and the model training may be overfitting.

Then, we investigate the constraint of intra-class irrelevant on the outputs of four blocks, the output of the last layer of the feature extraction network, and all five outputs, respectively. The results are shown in Appendix Table 8. We find that the average accuracy is improved by adding the IIC to the output of each module. However, when all the features are imposed on IIC at the same time, the performance is degraded. The possible reason is that imposing too many restrictions would affect the capability of recognition. Since the performance of applying IIC to the outputs of the four modules and the last layer has little difference, we prefer a more convenient way. If not specified, the following experiment applies IIC to the last layer output of the feature extraction network.

To further verify the accuracy gained brought by innovation, we also perform orthogonal decomposition on the features learned from the model with IIC to investigate the ability of face recognition of innovation. We provide the experimental results in Appendix Table 9. It shows that the learned innovation does have certain facial feature representation abilities. Combined with Tab 3, we can infer that the feature representation space obtained by IIC plays a role in performance improvement.

Finally, we validate the supposition mentioned in the Section 4.2. We use a small portion (i.e. 1/10) of MS1MV2 to train a ResNet50 as the backbone, with AdaFace serving as the basic face recognition algorithm. We provide the results in Appendix Table 10. It confirms our hypothesis that our proposed method can be regarded as a form of "feature augmentation".

## 5 LIMITATION

The method proposed in this paper is to find a more expressive feature representation space based on a pre-trained model. However, the method is currently applicable to face recognition tasks, and we think it has the potential to be extended to more general tasks and more general datasets. Though the improvement of recognition accuracy by IIC is greater than that listed in Table 2, it is not ruled out that a better feature expression space can be found by a suitable optimization algorithm.

## 6 CONCLUSIONS

Generally, traditional face recognition algorithms are to enhance FR performance by increasing intra-class correlation and inter-class incoherence. This paper does the opposite, and the proposed method further improves the accuracy of the face recognition algorithm by adding intra-class incoherence. Firstly, this paper analyzes the reasons for the improvement of face recognition accuracy from the feature representation level for different face recognition algorithms. Based on detailed analysis, we adopt the knowledge distillation framework to introduce innovation by adding intra-class irrelevant constraints (IIC), attempting to obtain a more representative feature space. Experiments show that the existing state-of-the-art method with IIC can be further improved, highlighting its potential to further enhance FR performance.

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

APPENDIX

Table 6: Comparison of recognition accuracy when the weight $\gamma$ of the intra-class uncorrelated constraint in Eq.2 is taken as 0.1, 0.5, 1.0, 5.0 and 10.0, respectively.

| Dateset | ArcFace | $\gamma$=0.1 | $\gamma$=0.5 | $\gamma$=1.0 | $\gamma$=5.0 | $\gamma$=10.0 |
|---|---|---|---|---|---|---|
| LFW | 99.15 | 99.43 | 99.33 | **99.50** | 99.43 | 99.40 |
| CFP | 95.69 | 96.16 | 96.23 | 96.03 | 96.06 | **96.31** |
| AgeDB | 94.35 | 94.37 | **94.47** | 94.30 | 94.28 | 94.25 |
| CALFW | 93.05 | 93.40 | 93.38 | 93.33 | 93.42 | **93.48** |
| CPLFW | 88.20 | 89.10 | 89.15 | **89.45** | 88.92 | 89.37 |
| VGG2 | 92.66 | **93.60** | 93.22 | 93.36 | 93.56 | 93.26 |
| AVG | 93.85 | 94.34 | 94.30 | 94.33 | 94.28 | **94.35** |

Table 7: Comparison of face recognition accuracy when the student network initialization mode is random initialization and teacher network weight initialization respectively, and the student network initialized with the teacher networks' weight does not using IIC, i.e., $\gamma$=0 in Eq.2.

| Dateset | ArcFace | Random Init. | Teacher weight Init. | w/o IIC |
|---|---|---|---|---|
| LFW | 99.15 | 99.37 | **99.50** | 99.22 |
| CFP | 95.69 | 95.57 | **96.03** | 94.84 |
| AgeDB | **94.35** | 93.68 | 94.30 | 93.62 |
| CALFW | 93.05 | **93.33** | **93.33** | 92.63 |
| CPLFW | 88.20 | 89.22 | **89.45** | 87.80 |
| VGG2 | 92.66 | 93.02 | **93.36** | 92.38 |
| AVG | 93.85 | 94.03 | **94.33** | 93.42 |

Table 8: Comparison of face recognition accuracy when IIC is applied to the outputs of four blocks, the output of the last layer (represented by 'F') of the feature extraction network respectively, the output of block 1 (represented by 'B.1') and the output of the last layer, the output of block 2 (represented by 'B.2') and the output of the last layer, and all five outputs (represented by 'overall '). In this setting, $\gamma$=0.5 .

| Dateset | ArcFace | B.1 | B.2 | B.3 | B.4 | F | B.1+F | B.2+F | overall |
|---|---|---|---|---|---|---|---|---|---|
| LFW | 99.15 | 99.40 | 99.43 | **99.45** | 99.40 | 99.33 | 99.40 | 99.36 | 99.22 |
| CFP | 95.69 | **96.57** | 96.31 | 96.03 | 96.07 | 96.23 | 96.43 | 95.69 | 95.20 |
| AgeDB | 94.35 | 94.18 | 94.25 | 94.13 | 94.12 | **94.47** | 94.35 | 93.87 | 93.73 |
| CALFW | 93.05 | 93.25 | **93.38** | **93.38** | 93.37 | **93.38** | 93.30 | 92.68 | 92.58 |
| CPLFW | 88.22 | 89.53 | **89.57** | 89.47 | 89.37 | 89.15 | 89.20 | 88.75 | 88.63 |
| VGG2 | 92.66 | 93.30 | 93.32 | 93.34 | 93.26 | 93.22 | **93.36** | 92.94 | 92.44 |
| AVG | 93.85 | 94.37 | **94.38** | 94.30 | 94.26 | 94.30 | 94.34 | 93.88 | 93.63 |

Table 9: Comparison of face recognition accuracy of innovation obtained from CosFace+IIC, Arc-Face+IIC, MagFace+IIC, AdaFace+IIC, and AdaFace*+IIC in the table. * represents the results of model training using the dataset MS1MV2.

| Dateset | CosFace | ArcFace | MagFace | AdaFace | AdaFace* |
|---------|---------|---------|---------|---------|----------|
| LFW     | 97.02   | 96.60   | 98.40   | 98.92   | 99.63    |
| CFP     | 85.16   | 86.06   | 91.39   | 94.56   | 95.24    |
| AgeDB   | 83.38   | 81.73   | 90.27   | 92.70   | 97.07    |
| CALFW   | 86.47   | 84.87   | 91.27   | 92.28   | 95.60    |
| CPLFW   | 80.48   | 80.53   | 84.08   | 87.00   | 90.35    |
| VGG2    | 83.30   | 84.98   | 88.20   | 92.60   | 93.68    |
| AVG     | 85.97   | 85.80   | 90.60   | 93.01   | 95.26    |

Table 10: Comparison of face recognition accuracy among some common and recent FR algorithms. All the models are trained with ResNet50 (He et al., 2016) as the backbone on CASIA.

| Method | Benchmarks(%) | | | | | | Average |
|--------|------|--------|---------|-------|-------|----------|---------|
|        | LFW  | CFP-FP | AgeDB-30 | CALFW | CPLFW | VGGFace2 |         |
| AdaFace | 99.40 | 92.86 | 94.92 | 94.67 | 88.73 | 92.64 | 93.87 |
| AdaFace+IIC | 99.48 | 93.57 | 94.98 | 94.40 | 88.73 | 93.22 | **94.07** |

