# OpenReview forum: "Enhanced Face Recognition using Intra-class Incoherence Constraint"
_ICLR.cc/2024/Conference — ICLR 2024 spotlight_

### Official Review · Reviewer_UpYx · 2023-10-31

**Soundness:** 3 good
**Presentation:** 3 good
**Contribution:** 3 good
**Rating:** 8
**Confidence:** 5

**Summary:**

This paper proposes a method to learn a new, better representation space for facial features, from two existing, already optimal representation spaces. The analysis is based on some geometrical consideration regarding the possibility to improve a representation feature space, by interpolating other representation spaces.
The feature augmentation approach demonstrates very limited improvements on some identification tests. However, the proposed idea has some merits and it is worth further considerations and discussion towards the direction of adaptive techniques for feature space augmentation.

**Strengths:**

Attempting to design a general feature space augmentation techniques interpolating already sub-optimal baseline spaces.

**Weaknesses:**

The reported results demonstrate a very limited improvement, maybe not justifying the efforts.
The designed feature augmentation model may be dependent on the training and produce different results on  different pre-trained models.
Language mistakes

**Questions:**

How would you expect to enlarge the scope of the proposed techniques?
How could you disentangle the proposed augmentation method from the training data, and consequently the learned features?

---

> ### Author Response · Authors · 2023-11-13
>
> Thank you for expressing your appreciation for our work and providing your insightful suggestions.
>
> Regarding Q.1, we acknowledge the limitation mentioned in our paper that our method has only been implemented on face recognition tasks. We are actively conducting further experiments in more challenging classification tasks to explore the potential application of our method beyond face recognition.
>
> Regarding Q2, we decompose the learned features into original features and innovation. We have compared the performance of the learned innovation across different models, as shown in Table 9 in the appendix of our paper.
>
> As for the weakness you mentioned, it is challenging to achieve significant accuracy improvements in face recognition, given the already high accuracy of existing benchmarks. As pointed out by Reviewer V2zT, “Existing face recognition benchmarks already have high accuracy, so the improvement doesn't look too significant.” We are actively exploring how to apply our approach in other domains.

---

### Official Review · Reviewer_GUup · 2023-10-31

**Soundness:** 3 good
**Presentation:** 3 good
**Contribution:** 3 good
**Rating:** 6
**Confidence:** 4

**Summary:**

This paper proposes a face representation learning method considering the intra-class dissimilarity. Experimental results on multiple datasets and baselines show the effectiveness of the proposed method.

**Strengths:**

This paper is well-written and easy to follow, the findings about pro-features and innovation features are interesting.

**Weaknesses:**

Here are some questions about this paper:
1) The loss L_{dissim} should be described more detailedly, especially the relationship between the motivation describe in introduction and minimizing the cosine similary of teacher and student feature.
2) The student network and teacher network are in the same sturcture. Will it be better if using the trained student network to be the teacher network since the goal of the method is a student network with higher performance? More analysis should be given.
3) Sicne this method mainly focuses on feature distillation and representation learning, more comparision with the SOTA distillation methods should be given.

**Questions:**

See weaknesses above

---

> ### Author Response · Authors · 2023-11-13
>
> Thank you for your valuable insights.
>
> For Weakness 1, we specify the computation as follows:
>
> $$L_{dissim}= \frac{<emb,ref\\_emb>}{||emb||_2\cdot||ref\\_emb||_2} $$
>
> Here, $<a,b>$ represents the inner product of vectors a and b, and $||\cdot||2$ denotes the L2 norm. The reason for this design is that as the cosine similarity of the vectors decreases, $L_{dissim}$ becomes smaller. This term is to make the two embeddings as dissimilar as possible, which intends to introduce innovation. In reality, the exact method for achieving the optimal solution is unknown. We aim to utilize the aforementioned dissimilarity measure between $emb$ and $ref\\_emb$ to promote learning better features. However, we acknowledge that there may be alternative approaches to describe this relationship more effectively.
>
> For Weakness 2, thank you for sharing your novel idea and we conduct some experiments following your idea（denoted as "ArcFace+IIC+IIC"）. The table shows the performance of different methods on various datasets.
>
> |  | LFW | CFP-FP | AgeDB-30| CALFW | CPLFW | VGGFace2 |Average
> | :----:| :----: | :----: | :----: | :----: | :----: | :----: | :----:
> | ArcFace+IIC | 99.43 | 96.03 | 94.12 | 93.05 | 89.45 | 93.36 | 94.24
> | ArcFace+IIC+IIC | 99.45 | 96.29 | 93.95 | 92.97 | 89.47 | 92.78 | 94.14
>
> Based on the results, it seems that the proposed method does not provide significant improvements. One possible reason for this could be that the optimal direction may be the unknown global optimum but our optimization is for an existing known feature. The proposed IIC method can improve the existing feature directions so that they become closer to local optima only. The use of this method multiple times may cause the feature direction to zigzag around the local optimal direction. As a result, the improvement could be limited.
>
> For Weakness 3, we appreciate your perspective on analyzing face recognition from the view of feature distillation and representation learning. However, most of the settings in state-of-the-art (SOTA) methods for distillation are different from ours. While our teacher model and student model share the same structure, this setting differs from the typical approach employed in the SOTA methods. Most SOTA methods aim to approximate the performance of the original model by reducing the model parameters, which often leads to a loss in accuracy.
>
> In contrast, our initial intention in designing the loss function was not solely focused on achieving distillation (i.e., reducing model parameters to closely match the performance of the original model). Instead, our goal is to learn new information and enhance model accuracy by leveraging the dissimilarity of features. Furthermore, we have compared our method with SOTA FR approaches and achieved better performance.

---

> > ### Comment · Reviewer_GUup · 2023-11-20
> > **Reply to the authors**
> >
> > Thank you for your detailed reply, and most of my concerns are addressed. Therefore, I have slightly increased my score.

---

> > > ### Author Response · Authors · 2023-11-20
> > >
> > > We sincerely appreciate your valuable suggestions and recognition of our work.

---

### Official Review · Reviewer_JUcW · 2023-10-31

**Soundness:** 3 good
**Presentation:** 2 fair
**Contribution:** 3 good
**Rating:** 6
**Confidence:** 4

**Summary:**

This paper starts with the hypothesis that when considering two face recognition (FR) models, the better-performing model can be further improved by combining features orthogonal to the worse-performing model. Authors propose the use of intra-class incoherence constraint (IIC) as a way of accomplishing this when a single FR model is available. Accuracy results on multiple face image datasets indicate that the proposed approach shows slightly better performance over other relevant approaches.

**Strengths:**

One strength of the paper is the observation that incorporating the features orthogonal to the inferior model can improve the performance of a superior model. I am still not convinced of this as there is no theoretical proof of this, but numerical results are reasonably convincing.

Another strength of the paper is the collection of experimental results. Authors show accuracy results on multiple face image datasets and show comparisons with the state of the art FR methods. Also, ablation studies included seem to support authors' hypothesis.

**Weaknesses:**

One major weakness of the paper is the most of the concepts and associated discussion seem to be based on the 2-D spaces in Figs. 1 and 2 whereas the feature vectors are in a much higher dimensional space. One problem with over-relying on the figures is that authors speak as if there is one perpendicular component to the inferior model in Fig. 1. For a given model in n-dimensional space, there is a (n-1)-dimensional space orthogonal to it. How do we choose the innovation feature from this space?

Another major weakness is that there are no theoretical proofs or justifications for the suggested improvements.  If a superior model can be further improved by combining it with something orthogonal to an inferior model, doesn't that imply that the "superior" model may not have been trained sufficiently? Also, why should we stop with combining the innovation from just one inferior model? Why not consider multiple inferior models and extract and use features orthogonal to these multiple models?

**Questions:**

1. Please clarify in the paper how the feature vector diagrams in Figs. 1 and 2 generalize to higher dimensions?

2. On Page 2, it is stated that "...innovation is always independent of features from CosFace". Independence and orthogonality are different concepts. Do you mean orthogonality or independence here? If later, please provide a justification for why orthogonality implies independence.

3. Manuscript suffers from language deficiencies. For example, "orthogonal" is a better choice for "perpendicular". Please revise the manuscript carefully to improve the language quality.

4. In Section 3.1, it is stated that "In the first case, as shown in Fig. 2(a), a and b are on different sides of d." In 2-D, it is clear what we mean by different sides. How are different sides defined in higher dimensional spaces?

5. In Eq. (3), should there be a summation over i on the RHS? If not, how come there is no dependence on i on the LHS?

**Details Of Ethics Concerns:**

No ethics concerns.

---

> ### Author Response · Authors · 2023-11-13
>
> Thank you very much for providing valuable feedback. For your questions, we shall address each of them separately.
>
>
> For Q.1, the feature vectors in Fig.1 and Fig.2 are used to illustrate the differences between the innovation and other feature vectors, as well as to demonstrate the possibility of synthesizing better features. As you mentioned, the high-dimensional space is more complex and difficult for visualization. Figs.1 and 2 are included to assist in showing the potential relationships among the features, which subsequently motivates the development of a new adaptive learning method based on IIC in this work to learn better features in high-dimensional space.
>
>
>
> For Q.2, our intention here is to state that innovation contains useful information for improving face recognition accuracy that was not yet captured by the inferior model. Innovation being orthogonal to the feature vector from the inferior model is consistent with this statement. We are going to use the term orthogonality, instead of independence, in the revised manuscript for clarity.
>
>
>
> For Q.3, thank you for your careful reading. We will make further improvements, particularly in the choice of terminology, and revise the manuscript carefully.
>
>
> For Q.4, we acknowledge that a and b being on different sides of d in fact implies the more general case where a and b are on different sides of the projection of d on the hyperplane specified by a and b. Under this scenario, the arguments presented in the manuscript remain valid. In other words, changing the modulus of the innovation may still be necessary to synthesize feature vectors close to the projection of the optimal feature vector d. We are going to point this out in the revised manuscript.
>
>
> For Q.5,  Eq. 3 and Eq. 4 are both total loss functions but they represent two different formulations. On the right hand side of Eq. 3, $L_{{dissim}_i}$ represents the dissimilarity between the outputs of the $i$-th layer of the teacher and student networks, where $i$ takes a value within [1, N]. In Eq. 4, we utilize all the available N+1 dissimilarities (N hidden layers + final output layer). As such, Eq. 3 can be considered as a special case of Eq. 4. We compare the performance under different loss functions  in Table 8 in the appendix of our paper. In the experiments, we chose to include the dissimilarity between the features in the final layer only.
>
> Furthermore, we would like to supplement some explanations regarding the concerns raised in your Weaknesses.
>
> For Weakness 1, we use the 2D space to describe the feature space solely for the purpose of convenience in analysis and visualization. Analyzing and synthesizing features in a high-dimensional space is a non-trivial task. It is precisely because we conduct thorough experiments and analysis in 2-D space that we come up with the idea of using innovation to improve model accuracy. Finally, we propose an adaptive learning method (i.e., IIC) in a high-dimensional space.
>
> For Weakness 2, we are trying to explore the theoretical proof for the suggested improvements. However, the fact that a superior model can be improved does not necessarily imply insufficient training but could also be due to the corresponding algorithm not effectively capturing more informative features. The idea of combining multiple inferior models is interesting, but how to properly integrate these features is a challenging task. This is because multiple models have different feature vectors, while the optimal vector is unknown. Nevertheless, we have made some attempts, and the experimental results are shown in the table below.
>
> |  | LFW | CFP-FP | AgeDB-30| CALFW | CPLFW | VGGFace2 |Average
> | :----:| :----: | :----: | :----: | :----: | :----: | :----: | :----:
> | 1 inferior model | 99.15 | 95.69 | 94.37 | 93.05 | 88.47 | 92.66 | 93.90
> | 2 inferior models| 99.10 | 95.61 | 94.05 | 93.03 | 88.45 | 92.44 | 93.78
>
> It seems that the method did not achieve significant improvements. One possible reason for this could be that the optimal direction is unknown and may represent a global optimum. However, our optimization is based on an existing known feature. Compared to the global optimal direction, the existing feature direction may be near the local optimal. The proposed IIC method can improve the existing feature directions to be closer to local optima. The use of multiple models may cause the feature direction to oscillate back and forth in the local optimal direction. As a result, the improvement is limited.

---

> ### Comment · Reviewer_JUcW · 2023-11-19
> **Upgraded rating**
>
> Authors, thank you for your careful responses to my review comments and questions and proposed revisions to your manuscript. Based on your response and other reviews, I have slightly increased my rating of your manuscript.

---

> > ### Author Response · Authors · 2023-11-19
> >
> > We deeply appreciate your recognition of our work :)

---

### Official Review · Reviewer_V2zT · 2023-11-01

**Soundness:** 3 good
**Presentation:** 4 excellent
**Contribution:** 3 good
**Rating:** 8
**Confidence:** 4

**Summary:**

In this paper, authors present a novel method that can improve existing face recognition (FR) methods by imposing a constraint of dissimilarity with learned embeddings.

The paper starts with a framework that decomposes the feature space of a superior model into two sub features, with one along that of a inferior model (pro-feature) and the other being orthogonal (innovation feature). Experiments show that the innovation part (orthogonal features)  has a high level of face distinguishability, which can be useful to learn from. Thus authors propose to use the dissimilarity with learning embedding as an auxiliary task in addition to the main face recognition tasks, under the knowledge distillation framework.

Experiments showed that this method can consistently improve existing methods (ArcFace, CosFace, MagFace, AdaFace) across 6 FR benchmarks. It's observed that the proposed method has a larger benefit on smaller datasets. Author hypothesized that the proposed method works as an feature augmentation mechanism.

**Strengths:**

The paper is well written and easy to follow.  The paper uses multiple sections to explain the main idea step by step. It starts with the introduction of feature decomposition and uses experiments to demonstrate the usefulness of the orthogonal subfeature; Then experiments show that moving the feature along the direction of the orthogonal subfeature can improve model performance; finally, authors  propose the idea to encourage the model to move towards the direction that dissimilar to learned embeddings.  The idea is implemented within the knowledge distillation framework.

The proposed method is novel and effective. Existing knowledge distillation methods generally impose a similarity objective to help the student model better mimic the teacher model. The proposed method is on the opposite direction - it leans dissimilarity.  Authors has provided a detailed analysis to justify this novel learning objective.

The experiments are extensive and the results are solid. Multiple classic face recognization models are used as baselines and studied on several common benchmarks. The improvements look consistent. Ablation studies provides good insights for understanding the proposed method.

**Weaknesses:**

The ablation of the proposed method can be further enhanced.  Authors has hypothesized the proposed method works as a feature augmentation mechanism. It will be insightful if other feature augmentation or regularization methods can be compared, for example, injecting noise to the activations.

The method is only studied on face recognition. In theory, this method can be applied to other classification tasks (especially fine-grained classification). Existing face recognition benchmarks already have high accuracy, so the improvement doesn't look too significant.

**Questions:**

It would be interesting if authors can conduct experiments on some more challenging classification tasks.

---

> ### Author Response · Authors · 2023-11-13
>
> Thank you very much for your recognition of our work and for providing us with promising suggestions.
> As stated in the limitations section in the paper, our method only proves to be effective in face recognition tasks, and its improvements for other classification tasks remain unknown. We are investigating its effectiveness in more challenging classification tasks to explore the potential application of our method beyond face recognition. We appreciate your valuable feedback and will continue to explore and expand the scope of our work.

---

### Meta-Review · Area_Chair_acFe · 2023-12-06

**Metareview:**

This paper proposes a novel method to incorporate the orthogonal features of inferior FR models to enhance FR performance. Their experiments and numerical results are convincing and detailed, together with proper ablation studies.  Concerns about writing as well as some details in the paper are well addressed in the rebuttal period, and all reviewers hold a positive attitude towards this submission. Therefore, I recommend acceptance.

**Justification For Why Not Higher Score:**

Even though experimental results are convincing, this paper lacks a theoretical explanation of its functionality.

**Justification For Why Not Lower Score:**

The method is novel with convincing experimental results.

---

### Decision · Program_Chairs · 2024-01-16

Accept (spotlight)